# Interactions of Clotrimazole with Certain *d*-Metal Compounds and with Organic Acids

Nina Skorik, Irina Kurzina *, Vladislav Korostelev, Dmitriy Fedorishin and Vladimir Kozik †

Faculty of Chemistry, National Research Tomsk State University, Tomsk 634050, Russia
* Correspondence: kurzina99@mail.ru; Tel.: +7-913-882-1028
† Passed away.

**Abstract:** During the interaction of aqueous-ethanol or ethanol solutions AgNO$_3$, H[AuCl$_4$], and CuCl$_2$, as well as aqueous suspensions of slightly soluble copper(II) salts Cu(C$_6$H$_4$NO$_2$)$_2$·H$_2$O, Cu(C$_7$H$_5$O$_2$)$_2$·3H$_2$O, and CuC$_7$H$_4$O$_3$·H$_2$O with the ethanol solution of clotrimazole at pH of ~(5.0–5.5), the [Ag(C$_{22}$H$_{17}$ClN$_2$)$_2$]NO$_3$·2H$_2$O, [Au(C$_{22}$H$_{17}$ClN$_2$)Cl$_3$], [Cu(C$_{22}$H$_{17}$ClN$_2$)$_2$Cl$_2$]·5H$_2$O, Cu(C$_{22}$H$_{17}$ClN$_2$)$_4$ (C$_6$H$_4$NO$_2$)$_2$, Cu(C$_{22}$H$_{17}$ClN$_2$)$_4$(C$_7$H$_5$O$_2$)$_2$, and Cu(C$_{22}$H$_{17}$ClN$_2$)$_3$(C$_7$H$_4$O$_3$)·2H$_2$O compounds are synthesised. They are characterised by elemental, thermal, thermogravimetric, and IR spectroscopic methods of analysis. The [Ag(C$_{22}$H$_{17}$ClN$_2$)$_2$]NO$_3$·2H$_2$O complex was shown to have a higher antimycotic activity against *Saccharomyces cerevisiae* fungi than that of AgNO$_3$ and C$_{22}$H$_{17}$ClN$_2$. Cocrystals/salts of the composition C$_{22}$H$_{17}$ClN$_2$·C$_6$H$_5$NO$_2$, C$_{22}$H$_{17}$ClN$_2$·C$_7$H$_6$O$_2$, 2C$_{22}$H$_{17}$ClN$_2$·C$_7$H$_6$O$_3$, and 2C$_{22}$H$_{17}$ClN$_2$·C$_{19}$H$_{19}$O$_6$N$_7$·H$_2$O are obtained from aqueous and aqueous ethanol suspensions containing nicotinic, benzoic, salicylic, and folic acids and clotrimazole (pH is 4.5–6.0). These cocrystals and salts were studied usin thermogravimetric, IR-spectroscopic methods. Diffraction patterns of the powders were obtained. The influence of the difference in the $pK_a$ components on the ability to form cocrystals/salts was assessed.

**Keywords:** synthesis; *d*-metals; aromatic and folic acids; clotrimazole; mixed-ligand salts; cocrystals/salts





## 1. Introduction

In recent years, interest in using metal compounds as antimicrobial/biocidal agents for combating infectious microorganisms has been revived [1]. Silver ions destroy pathogenic bacteria, viruses, and fungi, owing to their properties. Coordination compounds of copper-containing azoles are practically applied as biochemical and pharmacological preparations or catalysts obtained by various chemical processes. The authors of [2] point to the recent advances in the field of applying different metals and their complexes, used in biomedicine, in the diagnosis and treatment of chronic diseases. The methods used for investigating the synthesised biologically active complexes of Mn$^{2+}$, Co$^{2+}$, Ni$^{2+}$, and Fe$^{3+}$ are being expanded; their antibacterial, antimicrobial, and anticancer activities are being investigated, including the ionic radii influence of the metals on the biological activity of drugs [3]. The synergetic effects of mononuclear transition metal complexes containing mixed ligands are being investigated in terms of their antimicrobial and antioxidant activities [4].

One of the azoles (C$_{22}$H$_{17}$ClN$_2$ (Clm) clotrimazole) produces antibacterial, antiprotozoal, trichomonicidal, and antimycotic pharmacological pluripotential effects; the preparation possesses anticancer properties. The individual and combined effects produced by substances, such as clotrimazole and the salts of biogenic elements ZnSO$_4$, CuSO$_4$, AgNO$_3$, and NiSO$_4$, on microbial cultures [5], and the antitumor properties of clotrimazole complexes containing metals (palladium, ruthenium, platinum, and silver) have been researched. The silver complex containing clotrimazole has been found to be nontoxic for mammalian cells and is an effective nanoantibiotic [6]. A new palladium complex [Pd(Clm)$_2$Cl$_2$], synthesised by the reaction of the interaction between bis(acetonitrile)palladium dichloride

PdCl$_2$(CH$_3$CN)$_2$ and clotrimazole [7], demonstrates increased cytotoxicity in relation to tumour cells as compared to clotrimazole itself. A series of twelve compounds, obtained in the systems Ru–KTZ and Ru–Clm (where KTZ is ketoconazole, C$_{26}$H$_{28}$Cl$_2$N$_4$O$_4$), for instance, Ru(KTZ)$_2$Cl$_2$, Ru(Clm)$_2$Cl$_2$, has been shown to possess anticancer properties [8]. New transplatinum(II) complexes of the composition [Pt(Clm)$_2$I$_2$] and [Pt(Clm)$_2$Cl$_2$], respectively, have been obtained and characterised by the reaction of the interaction of K$_2$[PtCl$_4$] with KI and Clm, as well as K$_2$[PtCl$_4$ ] with Clm [9]. These complexes of platinum(II) have inhibited tumour cells without cytotoxicity signs. Sixteen new compounds of cobalt(II), nickel(II), zinc(II), and copper(II) containing clotrimazole, and showing cytotoxic activity, have been described in [10]: [M(Clm)$_2$Cl$_2$]·$n$H$_2$O, [M(Clm)$_2$Br$_2$]·$n$H$_2$O, [M(Clm)$_3$Br$_2$], [M(Clm)$_3$NO$_3$]NO$_3$·$n$H$_2$O, [M(Clm)$_3$(NO$_3$)$_2$]·$n$H$_2$O, [M(Clm)$_3$(H$_2$O)$_2$NO$_3$]NO$_3$·$n$H$_2$O, and others. In [11], the synthesis and characteristics of the complexes of metals containing clotrimazole and manifesting activity against the pathogenic agent of the Chagas disease have been presented: [AuClmCl$_3$], K$_2$[PtCl$_4$(Clm)$_2$], and [Cu(Clm)$_2$](PF$_6$), which were obtained when Clm interacted with H[AuCl$_4$], K$_2$[PtCl$_4$] and with tetrakis(acetonitrile)copper(I) hexafluorophosphate [Cu(CH$_3$CN)$_4$](PF$_6$), respectively. All new complexes were characterised by means of NMR and other methods. A square arrangement of the nitrogen atom of the Clm molecule and three chloride atoms around the aurum(III) ion can be observed in the [AuClmCl$_3$] complex. In [12] the synthesis and characteristics of new complexes of copper(II) and aurum(I) containing clotrimazole and ketoconazole have been described. The compounds [Cu(Clm)$_4$]Cl$_2$·2H$_2$O, [Cu(Clm)Cl$_2$]$_2$, [Cu(KTZ)$_3$Cl$_2$], and [Cu(KTZ)Cl$_2$]$_2$·2H$_2$O have been obtained when CuCl$_2$ interacted with Clm and KTZ in acetonitrile. The complexes of aurum(I) [Au(PPh$_3$)(Clm)](PF$_6$) and [Au(PPh$_3$)(KTZ)](PF$_6$)·H$_2$O have resulted from the interaction of triphenylphosphine aurum(I) chloride Au(PPh$_3$)Cl ((P(C$_6$H$_5$)$_3$, PPh$_3$ is triphenylphosphine) with K(PF$_6$) and Clm or KTZ in acetonitrile. The [Cu(Clm)$_4$]Cl$_2$·2H$_2$O complex has a square-shaped flat structure, typical of the complexes of copper(II) tetrakis-imidazole. The authors of [13], by the example of the *Candidaalbicans* culture, have shown that there is a synergism of the action of silver ions and clotrimazole. The synthesis of nucleic acids, lipids, and polysaccharides in the cells of a pathogenic fungus is inhibited under the influence of clotrimazole. The authors of [14] have shown that, in vivo, the ruthenium–clotrimazole complex influences significantly a mouse model of cutaneous leishmaniasis, yielding imperceptible toxicity with respect to normal cells of mammals. Three new ruthenium(II)–clotrimazole complexes, containing diphosphine ligands and having an antimicrobial effect on Mycobacterium tuberculosis, have been synthesised and characterised [15]. Three neutral cyclometallic platinum(II) complexes, containing imidazolyl derivatives of 1-methylimidazole (CH$_3$C$_3$H$_3$N$_2$, MeIm) and antifungal drugs, such as clotrimazole (Clm) and bifonazole (C$_{22}$H$_{18}$N$_2$, BFZ), have been synthesised and characterised [16]. The [PtMeIm] and [PtBFZ] complexes exhibit higher cytotoxicity than cisplatin does. The therapeutic activity of clotrimazole, whose application is being expanded, can be strengthened, not only by synthesising new coordination compounds based on it and metal ions, but also by combining it with other components. So, for example, an ointment based on chitosan, clotrimazole nanoparticles, and a juice extract of the Egyptian grape "Vitis vinifera" can be used as a new anti-dermatophyte agent having a high wound-healing effect [17].

In the last decade, the pharmaceutical use of cocrystals/salts has been of great interest. The scientific community has been in discussion about the differences between a molecular salt and a cocrystal [18]. A solid system is called a molecular salt, provided that a proton travels from the acid to the base and the components are ionized, but in a cocrystal, the components are in a neutral form and ionic interactions are absent. However, a certain solid phase cannot always be unequivocally considered to be a molecular salt or cocrystal; (one and the same component can be present in the crystalline lattice in both ionized and neutral forms). Therefore, for example, highly conductive imidazolium salts known as phthalate and terephthalate of imidazolium have been obtained [19]. The conductivity is associated with the proton jumps occurring in the acid–base pairs through the bridges formed by

water molecules [20]. In [21,22], the problem of the so-called salt–crystalline continuum, related to the formation of a cocrystal or salt, depending on the $\Delta pK_a$ of the interacting acid and base, is discussed. The reaction between the acid and the base is generally accepted to produce a salt if the $\Delta pK_a$ ($pK_a$ (base) $-$ $pK_a$ (acid)) is greater than 2 or 3; this criterion is frequently used when choosing counter-ions while synthesising the salt. When the $\Delta pK_a < 3.75$, the COOH·N interaction is also supposed to take place, accompanied by the formation of cocrystals; whereas, if the $\Delta pK_a > 3.75$, the proton is transported from the HL acid to the B base and the $(HB)^+(L)^-$ salt is formed. In the above-mentioned works, the formation of cocrystals having $\Delta pK_a$ values in the range of $(-1)$–2 is demonstrated, and the salts are formed at values of $pK_a > 3.4$. However, it is specified that in the range of $\Delta pK_a$ 0–3, there is ambiguity. Forecasts, made on the basis of the selected assessments, do not always agree with the experimental data.

Clotrimazole $C_{22}H_{17}ClN_2$ is a representative of the aromatic series of heterocycles:

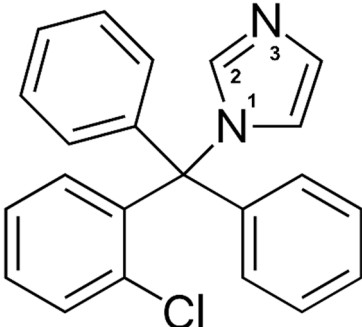

As a derivative of imidazole, it can manifest base properties (nitrogen base), protonated by the lone-atom electron pair $N_{(3)}$ ($lgB_1 = 5.99$ [23], where $B_1$ is a protonization Clm constant involving the formation of a clotrimazole $HClm^+$ particle). The presence of the donor $N_{(3)}$ atom also promotes the binding of clotrimazole with ions of transition metals, such as platinum, ruthenium, palladium, copper, silver, cobalt, etc.

The purpose of this work has been to synthesise and study the physico–chemical and antimycotic ($[Ag(C_{22}H_{17}ClN_2)_2]NO_3 \cdot 2H_2O$) properties of the compounds of clotrimazole containing the salts of silver (nitrate), gold (tetrachloroaurate), and copper (chloride, nicotinate, benzoate, and salicylate), as well as compounds of clotrimazole, including nicotinic, benzoic, salicylic, and folic acids.

## 2. Objects and Methods

The thermal stability of the synthesised compounds was studied using the NETZSCHSTA 449 C device (NETZSCH, Berlin, Germany). The electronic spectra of the solution absorption were recorded by the LEKISS 2107UV spectrophotometer. The pH of the solutions was measured using a pH meter of the pH-673 brand (The Preston Hire, Sydney, Australia), whose glass electrode was calibrated according to buffer solutions having a pH ranging from 3.56 to 6.86. The IR spectra of the salts in the tablets made from KBr were recorded using the ThermoNicollet NEXUS FTIR spectrometer (Thermo Fisher, New York, NY, USA) in the frequency range of 4000–400 $cm^{-1}$. The synthesised compounds were studied by automatic elemental CHNS analysis by means of the EURO EA 3000 analyser (Eurovector, Pavia, Italy), applying the Sartorius MSE 3.6P-000-DM microbalance (Sartorius, Göttingen, Germany). The XPA was made using the SHIMADZU XRD 6000 X-ray diffractometer (SHIMADZU, Moscow, Russian) equipped with the Cu X-ray tube. The antimycotic activity of the $[Ag(C_{22}H_{17}ClN_2)_2]NO_3 \cdot 2H_2O$ complex was studied by the well diffusion method in the standard Saburo medium with the authors' modification. Clotrimazole was purchased from Sigma (Anseong-si, Republic of Korea); metal salts and organic acids were marked as "chemically pure" or "pure for analysis". All the reagents and solvents were used without supplementary purification.

### 3. Experimental

#### 3.1. Synthesis of Mixed-Ligand Salts $[AgClm_2]NO_3 \cdot H_2O$, $[AuClmCl_3]$, $[CuClm_2Cl_2] \cdot 5H_2O$

Nitrate of bis-clotrimazoleargentum(I) obtained from aqueous ethanol solutions having a mole ratio of the components equal to 1:2 was synthesised in accordance with the reaction equation:

$$AgNO_3 + 2Clm = [AgClm_2]NO_3$$

A sample containing 0.038 g of $AgNO_3$ was dissolved in the mixture consisting of 1 mL of water and 3 mL of ethyl alcohol to obtain ~0.2 g of the product. This clear solution was mixed with 3 mL of ethanol, containing 0.154 g of clotrimazole. The pH of the mixture was 5.6; (the glass electrode was calibrated according to alcohol–water HCl solutions of the known concentration). Then, the mixture was kept in darkness for several days, when the formation of the crystalline, slightly soluble precipitation $[Ag(C_{22}H_{17}ClN_2)_2]NO_3 \cdot 2H_2O$ of dark brown colour was observed. This precipitation was rinsed with alcohol and dried in the air. The product yield was 90%.

The *Saccharomyces cerevisiae* fungi were used as a test system to research the antimycotic activity of the $[Ag(C_{22}H_{17}ClN_2)_2]NO_3 \cdot 2H_2O$ complex [24]. The system incubation (30 °C, 24 h) was followed by measuring the halo of inhibition of the east fungi growth with up to 0.1 mm accuracy. The "blank experiments" involved preparations of clotrimazole and silver nitrate whose content corresponded to their content in the complex under study (Figure 1).

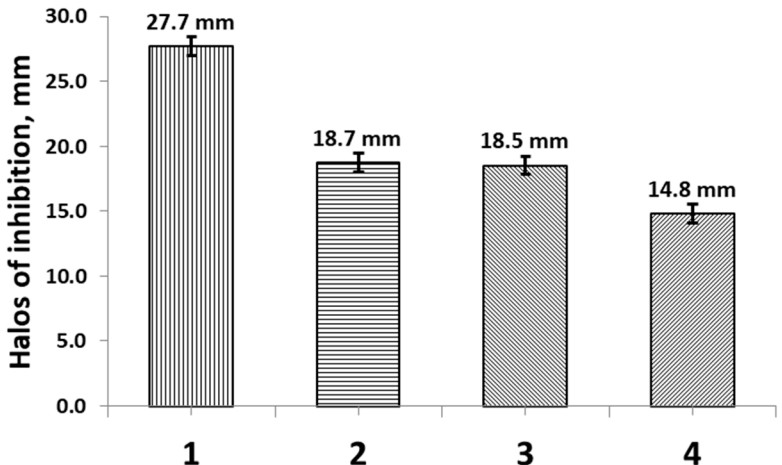

**Figure 1.** The size of halos of inhibited growth of *Saccharomyces cerevisiae* containing the following compounds: 1—$[Ag(C_{22}H_{17}ClN_2)_2]NO_3 \cdot 2H_2O$, 5 mg/mL; 2—$C_{22}H_{17}ClN_2$, 5 mg/mL; 3—$C_{22}H_{17}ClN_2$, 3.85 mg/mL; 4—$AgNO_3$, 0.95 mg/mL.

Tris-chloroclotrimazoleaurum(III) was obtained from the aqueous ethanol solution by the reaction:

$$H[AuCl_4] + Clm = [AuClmCl_3] + HCl$$

The $H[AuCl_4]$ aqueous ethanol solution and the ethanol Clm solution were taken at a mole ratio of the components equal to 1:1; the mixture pH was ~4. Three days later, the light yellow precipitate was rinsed with alcohol and dried in the air. The product yield was 84%. The results of the elemental, gravi-, and thermogravimetric analyses of mixed-ligand salts (MLS) of argentum(I) and aurum(III) are presented in Tables 1 and 2. The synthesis of the bis-clotrimazolecopper(II) chloride from ethanol or aqueous–ethanol solutions at a mole ratio of the components of 1:2 is presented by the reaction:

$$CuCl_2 + 2Clm = CuClm_2Cl_2$$

A salt $CuCl_2 \cdot 2H_2O$ sample weighing 0.05 g in 3 mL of ethyl alcohol was mixed with 5 mL of the alcoholic solution containing clotrimazole (0.2 g). Every other day, the formed greenish yellow amorphous precipitate was filtered, rinsed with alcohol, dried in the air, and subjected to elemental analysis (Table 1).

**Table 1.** Analytical data of biligand salts of argentum(I), aurum(III), and copper(II) containing clotrimazole.

| Compound | N, % | | C, % | | H, % | | Ag, Au, CuO, % | | $H_2O$, % | |
|---|---|---|---|---|---|---|---|---|---|---|
| | f* | c* | f | c | f | c | f | c | f | c |
| $[Ag(C_{22}H_{17}ClN_2)_2]NO_3 \cdot 2H_2O$ | 8.5 | 7.82 | 59.1 | 58.96 | 4.2 | 4.24 | 11.7 | 12.04 | 4.2 | 4.02 |
| $[Au(C_{22}H_{17}ClN_2)Cl_3]$ | 5.2 | 4.32 | 40.5 | 40.73 | 2.8 | 2.62 | 28.3 | 30.39 | – | – |
| $Cu(C_{22}H_{17}ClN_2)_2Cl_2 \cdot 5H_2O$ | 5.9 | 6.13 | 54.9 | 57.81 | 4.1 | 4.85 | 9.0 | 8.70 | – | – |
| $Cu(C_{22}H_{17}ClN_2)_4(C_6H_4NO_2)_2$ | 7.9 | 8.29 | 68.7 | 71.13 | 5.2 | 4.50 | 4.7 | 4.71 | – | – |
| $Cu(C_{22}H_{17}ClN_2)_4(C_7H_5O_2)_2$ | 6.7 | 6.65 | 72.4 | 72.63 | 4.9 | 4.63 | 4.2 | 4.72 | – | – |
| $Cu(C_{22}H_{17}ClN_2)_3 (C_7H_4O_3) \cdot 2H_2O$ | 5.9 | 6.61 | 65.8 | 68.96 | 4.1 | 4.64 | 6.5 | 6.26 | 2.7 | 2.83 |

f*, c*—found, calculated.

**Table 2.** Analysis of thermograms of mono- and biligand salts of *d*-metals and cocrystals/salts of organic acids and clotrimazole.

| No. | Nature of Effect | Temperature Interval, °C | Mass Loss (Compared to Init.), Residue, % | | Corresponding Process |
|---|---|---|---|---|---|
| | | | f | c | |
| | | | $Cu(C_7H_5O_2)_2 \cdot 3H_2O$ | | |
| 1 | Group of endo-effects | 40–130 | 15.5 | 15.01 | Water loss |
| 2 | Exo-effect | 130–450 | 65.2 | 67.33 | Destruction of benzoate-ion |
| 3 | Endo-effect | 450–900 | 22.0 | 22.11 | Formation of CuO oxide |
| | | | $[Ag(C_{22}H_{17}ClN_2)_2]NO_3 \cdot 2H_2O$ | | |
| 1 | Endo-effect | 25–160 | 4.0 | 4.02 | Water loss |
| 2 | Exo-effects | 216–410 | 46.7 | 45.43 | Decomposition of a nitrate-ion and loss of Clm |
| 3 | Exo-effect | 410–900 | 39.3<br>10.6 | 38.5<br>12.05 | Combustion of Clm, formation of Ag |
| | | | $[Au(C_{22}H_{17}ClN_2)Cl_3]$ | | |
| 1 | Group of endo-, exo-effects | 171–280 | 36.1 | 34.90 | Elimination of 0.5Clm + 0.5(1.5Cl$_2$) |
| 2 | Exo-effect | 410–900 | 34.5<br>28.3 | 34.90<br>30.38 | Elimination of 0.5Clm + 0.5(1.5Cl$_2$); formation of Au |
| | | | $C_{22}H_{17}ClN_2 \cdot C_6H_5NO_2$ | | |
| | Endo-effect | 119.3 | 0.0 | 0.0 | Melting |
| | Endo-effects | 199–400 | 72.2 | 73.69 | Loss of Clm |
| | Exo-effect | 400–800 | 27.8 | 26.31 | Loss of HNic |
| | | | $2C_{22}H_{17}ClN_2 \cdot C_{19}H_{19}O_6N_7 \cdot H_2O$ | | |
| | Endo-effects | 120–180 | 1.5 | 1.57 | Loss of $H_2O$ |
| | Endo-effect | 230–390 | 60.0 | 60.02 | Loss of 2Clm |
| | Exo-effect | 390–800 | 39.4 | 38.41 | Loss of $H_2Fol$ |

The electronic absorption spectra of the systems, where mixed-ligand complexation occurs, can indirectly confirm the process of isolating the solid mixed-ligand salts (MLS). So, for example, the electronic absorption spectra of alcoholic solutions, containing copper(II) chloride and Clm, have the highest optical density as compared to that of the solutions

of separate components of the same concentration, which reveals the CuCl$_2$–Clm system formation having a new compound: a mixed-ligand complex (Figure 2).

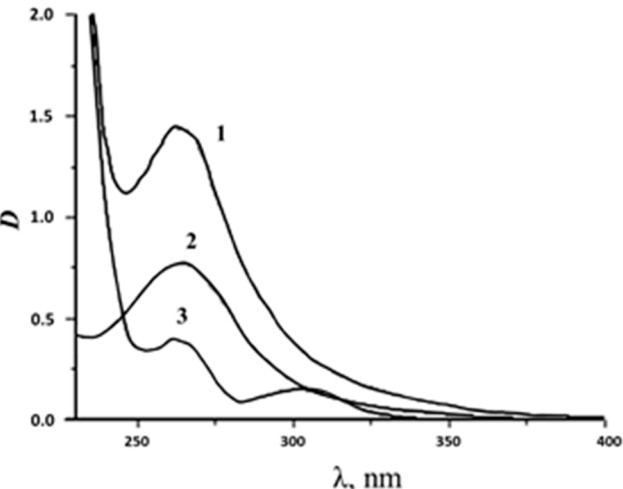

**Figure 2.** Electronic absorption spectra of the systems in ethanol: 1—CuCl$_2$–Clm ($C_{Cu}$ = $C_{Clm}$ = 5 × 10$^{-4}$ mol/L, pH = 4.7, $\lambda_{max}$ = 262 nm); 2—CuCl$_2$ ($C_{Cu}$ = 5 × 10$^{-4}$ mol/L, pH = 3.8, $\lambda_{max}$ = 264 nm); 3—Clm ($C_{Clm}$ = 5 × 10$^{-4}$ mol/L, pH = 5.7, $\lambda_{max}$ = 261 nm).

*3.2. Synthesis of Nicotinate* Cu(C$_6$H$_4$NO$_2$)$_2$·H$_2$O, *Benzoate* Cu(C$_7$H$_5$O$_2$)$_2$·3H$_2$O, *and Salicylate* CuC$_7$H$_4$O$_3$·H$_2$O *of Copper(II)*

The initial salts of copper(II): benzoate Cu(C$_7$H$_5$O$_2$)$_2$·3H$_2$O (Cu(Benz)$_2$·3H$_2$O), nicotinate Cu(C$_6$H$_4$NO$_2$)$_2$·H$_2$O (Cu(Nic)$_2$·H$_2$O), and salicylate CuC$_7$H$_4$O$_3$·H$_2$O (CuSal·H$_2$O), required for synthesising mixed-ligand salts involving clotrimazole, were obtained by the reaction of the interaction between copper(II) dichloride containing aromatic acids, neutralised by the sodium hydroxide:

$$CuCl_2 + ((2, 1)H_{1\text{-}2}L + 1.8NaOH) \rightarrow CuL_{2\text{-}1} \downarrow$$

The final pH value of the mixture equal to 4.7–4.9 was obtained by using the NaOH and HCl solutions. When crystallised, the salt was then filtered, rinsed with cold water, removing chloride ions, and dried in the air. The crystallisation water content was determined in the salts by heating them at a temperature of 125 °C for 2 h, and the CuO oxide content was detected by calcination at 900 °C (Table 3). The content of the salts was also confirmed by the thermogravimetric method data; (for example, the results of the thermogravimetric analysis of copper(II) benzoate are provided in Table 2). Monoligand salts represent coloured substances that are slightly soluble in water. The solubility constants $K_S$ of benzoate and copper(II) salicylate, which we determined according to the data on the salt solubility in the HCl solutions of different concentrations, are 1.66 × 10$^{-7}$ (ionic strength is $I$ = 0.1) and 2.45 × 10$^{-14}$ ($I$ = 0.3), respectively. In the case of copper(II) nicotinate, $K_S$ = 1.66 × 10$^{-10}$ ($I$ = 0.3) [25].

**Table 3.** Data of the thermal analysis of the monoligand copper(II) salts.

| Compound | CuO, % | | H$_2$O, % | |
|---|---|---|---|---|
| | f | c | f | c |
| Cu(C$_6$H$_4$NO$_2$)$_2$·H$_2$O | 24.5 | 24.42 | 5.1 | 5.53 |
| CuC$_7$H$_4$O$_3$·H$_2$O | 35.7 | 36.54 | 8.3 | 8.27 |
| Cu(C$_7$H$_5$O$_2$)$_2$·3H$_2$O | 22.1 | 22.11 | 15.7 | 15.01 |

*3.3. Syntheses of the Mixed-Ligand Copper(II) Salts of the Composition*
$Cu(C_{22}H_{17}ClN_2)_4(C_6H_4NO_2)_2$ *(CuClm$_4$Nic$_2$),* $Cu(C_{22}H_{17}ClN_2)_4(C_7H_5O_2)_2$
*(CuClm$_4$Benz$_2$),* $Cu(C_{22}H_{17}ClN_2)_3(C_7H_4O_3)·2H_2O$ *(CuClm$_3$Sal·2H$_2$O)*

The mixed-ligand salts of copper(II), containing an aromatic acid anion and neutral Clm molecules as ligands, were synthesised from aqueous–alcoholic suspensions of slightly soluble benzoate, nicotinate, and salicylate of copper(II) and azole at pH of ~5.5 at different mole ratios of the initial components by the reaction:

$$CuL_{2\text{-}1(susp)} + xClm_{(s)} \rightarrow CuClm_xL_{2\text{-}1} \downarrow,$$

where L is an anion of the aromatic acid $Benz^-$, $Nic^-$, $Sal^{2-}$.

Nicotinate of tetrakis-clotrimazolecopper(II) $[Cu(C_{22}H_{17}ClN_2)_4](C_6H_4NO_2)_2$ (CuClm$_4$Nic$_2$) was synthesised by the reaction:

$$Cu(C_6H_4\,NO_2)_{2(s)} + 5C_{22}H_{17}ClN_2 \rightarrow [Cu(C_{22}H_{17}ClN_2)_4](C_6H_4NO_2)_2\downarrow$$

It was synthesised from the aqueous–alcoholic solution by the interaction of a slightly soluble copper(II) nicotinate salt $CuNic_2·H_2O$ containing clotrimazole at a molar ratio of the components of 1:5 and pH = 5.6. For this purpose, 3 mL of the clotrimazole (0.12 g) alcoholic solution was mixed with the aqueous suspension of copper(II) nicotinate (0.025 g of $CuNic_2·H_2O$, 3 mL of $H_2O$). Over the course of 7 days, copper(II) nicotinate transformed to a new phase, changing the colour. The isolated compound was filtered, rinsed with ether, and dried in the air. Benzoate of tetrakis-clotrimazolecopper(II) $[Cu(C_{22}H_{17}ClN_2)_4](C_7H_5O_2)_2$ ([CuClm$_4$]Benz$_2$) was synthesised in the same way as nicotinate of tetrakis-clotrimazolecopper(II). The salicylate of tris-clotrimazolecopper(II) $Cu(C_{22}H_{17}ClN_2)_3C_7H_4O_3·2H_2O$ was synthesised by the reaction:

$$CuC_7H_4O_{3(s)} + 5C_{22}H_{17}ClN_2 \rightarrow Cu(C_{22}H_{17}ClN_2)_3C_7H_4O_3 \downarrow$$

It was synthesised from the aqueous–alcoholic solution during the interaction between the slightly soluble salt $CuSal·H_2O$ and clotrimazole at pH of 5.2. To achieve this, 3 mL of the clotrimazole alcoholic solution was mixed with the aqueous suspension of copper(II) salicylate (3 mL of $H_2O$), creating a mole ratio of CuSal:Clm = 1:5 in the mixture. Over the course of 7 days, the copper(II) salicylate transformed into a new phase, changing the colour. The new compound was filtered, rinsed with ether, and dried in the air. The composition of the mentioned biligand salts of copper(II) was established by means of elemental, gravimetric methods of analysis (Table 1).

*3.4. Syntheses and Properties of Cocrystals/Salts of the $C_{22}H_{17}ClN_2·C_6H_5NO_2$ (Clm·HNic),*
$C_{22}H_{17}ClN_2·C_7H_6O_2$ *(Clm·HBenz),* $2C_{22}H_{17}ClN_2·C_7H_6O_3$ *(2Clm·H$_2$Sal),*
$2C_{22}H_{17}ClN_2·C_{19}H_{19}O_6N_7·H_2O$ *(2Clm·H$_2$Fol·H$_2$O)·Composition*

In this work, cocrystals/salts were synthesised from aqueous or aqueous–alcoholic suspensions, containing aromatic acids (nicotinic, benzoic, and salicylic), folic acid, and clotrimazole. Long-term holding of the suspensions leads to obtaining a new solid phase:

$$(1\text{-}2)Clm + H_{1\text{-}2}L = (1\text{-}2)Clm·H_{(1\text{-}2)}L,$$

accompanied by its subsequent filtration and rinsing with ether.

The compound Clm·HNic was synthesised by the introduction of a dry sample of clotrimazole (0.12 g) into the aqueous suspension (4 mL of $H_2O$) of nicotinic acid (0.025 g) at a mole ratio of the components of Clm:HNic = 2:1. The $pH_{mixture}$ was ~4.5, there was long-term holding of the mixture, bath heating at ~70 °C (for ~10 min), cooling, filtration of a new phase, and rinsing with ether. The Clm·HBenz compound was obtained using

the same method. The cocrystal/salt of clotrimazole containing the salicylic acid was synthesised by the reaction:

$$2Clm + H_2Sal = 2Clm \cdot H_2Sal$$

The amounts of 0.1 and 0.15 g of dry clotrimazole were added, while stirring, to a weighed portion of 0.05 g of the salicylic acid into 4 mL of water (the precipitate swells in this case). While stirring, the mixture was held for ~4 min in a water bath at 60 °C (subsequent heating solidifies the precipitate), then the mixture was held in the air for several days (the suspension pH was 4.5). The precipitate was filtered, rinsed with ether, and dried in the air.

The $2C_{22}H_{17}ClN_2 \cdot C_{19}H_{19}O_6N_7 \cdot H_2O$ cocrystal/salt at a mole ratio of $Clm:H_2Fol = 2.7:1$ was synthesised from the aqueous–alcoholic solution by pouring the suspension containing 0.07 g of $H_2Fol \cdot 2H_2O$ into 4 mL of $H_2O$ along with 5 mL of the alcoholic solution and 0.15 g of Clm (the mixture pH was 5.9). Two days later, the precipitate was rinsed with ethanol and dried in the air; (the yield was 63%). The elemental analysis data of the synthesised cocrystals/salts are in Table 4.

**Table 4.** Analytical data of cocrystals/salts of benzoic, nicotinic, salicylic, and folic acids ($H_{(1-2)}L$) with clotrimazole.

| Compound | N, % | | C, % | | H, % | | Clm, % | | $H_{1-2}L$, % | |
|---|---|---|---|---|---|---|---|---|---|---|
| | f | c | f | c | f | c | f | c | f | c |
| $C_{22}H_{17}ClN_2 \cdot C_7H_6O_2$ | 7.6 | 6.00 | 75.9 | 74.52 | 5.3 | 4.93 | – | – | – | – |
| $C_{22}H_{17}ClN_2 \cdot C_6H_5NO_2$ | 8.4 | 8.98 | 75.5 | 71.80 | 5.3 | 4.70 | 72.2 | 73.69 | 27.8 | 26.31 |
| $2C_{22}H_{17}ClN_2 \cdot C_{19}H_{19}O_6N_7 \cdot H_2O$ | 12.0 | 13.48 | 66.0 | 65.8 | 4.8 | 4.79 | 60.0 | 60.02 | 39.4 | 38.41 |
| $2C_{22}H_{17}ClN_2 \cdot C_7H_6O_3$ | 7.3 | 6.76 | 74.7 | 73.93 | 4.9 | 4.83 | 84.7 | 83.32 | 14.8 | 16.68 |

## 4. Results and Discussion

The synthesised mixed-ligand salts of d-metals and cocrystals/salts were analysed by elemental (Tables 1 and 4), thermal (Table 1), and thermogravimetric methods (Tables 2 and 4). The thermogravimetric analysis data were used to determine the quantitative content of water, ligand, and metal (or metal oxide) in the salts of metals and to define the content of acids and clotrimazole in cocrystals/salts since, as Table 2 shows, the processes of loss, destruction, and formation of the mentioned components of the synthesised compounds are in different temperature ranges. The data on elemental analysis of cocrystals/salts (Table 4) are quantitatively confirmed by the thermogravimetric method (Table 2). Thermogravimetric studies of the synthesised compounds also allow for the suggestion of a mechanism for their thermal decomposition. These studies are also vital for understanding the thermal stability of the salts, which, along with other properties, is a characteristic of chemical compounds.

Clotrimazole is included in the composition of the synthesised salts of argentum(I), aurum(III), and copper(II) in the form of a neutral molecule. The diagram of the distribution of clotrimazole Clm molecules and the product of their protonisation ($HClm^+$), depending on the medium pH (Figure 3) when the pH of the salt synthesis is ~5.5, shows that there are protonated and neutral (~(65, 35)%) particles of clotrimazole, respectively.

The competitive response of clotrimazole protonated particles containing a metal ion:

$$HClm^+ + M^{n+} \leftrightarrow MClm^{n+} + H^+$$

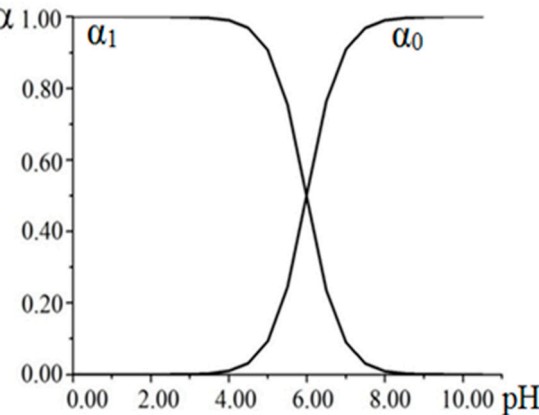

**Figure 3.** Distribution diagram of clotrimazole particles, depending on the solution pH: $\alpha_1$—HClm$^+$, $\alpha_0$—Clm ($C_{Clm}$ = 0.01 mol/L, lg$B_1$ = 5.99).

This leads to the formation of a neutral Clm molecule, whose pyridinic atom $N_{(3)}$ participates in the reaction of forming a complex with the metal ion, since it contains unsplit pair electrons on the $sp^2$-hybrid orbital. This is confirmed by studying IR spectra of clotrimazole compounds. The authors of [26] point to the frequency shift and intensity change of the characteristic peak of the clotrimazole C=N binding from 1436 cm$^-$ to 1489 cm$^-$ and 1444 cm$^-$ in the IR spectra of its complexes, accompanied by the charge transfer with $\pi$-acceptors—tetracyanoethylene and 7,7′, 8,8′-tetracyanoquinodimethane. They associate these phenomena with the participation of the azole C=N group in the complex formation involving acceptors. In [27], the shift in the stretching bands 1495 and 1447 cm$^-$ of the heterocycle C=N bond during the complex formation of imidazole and copper(II) chloride is mentioned.

To prove the synthesis of new salts, participation of clotrimazole in the MLS formation, IR spectra of absorption of initial components, and the products of their interaction have been analysed in this work. The indication of the metal ion coordination with the base nitrogen atom $N_{(3)}$ of clotrimazole can be a shift of its stretching band of the heterocycle bond C=N (1436.9 cm$^-$) to the high-frequency range in salts: [Ag($C_{22}H_{17}ClN_2$)$_2$]NO$_3$·2H$_2$O–1442.6, [Au($C_{22}H_{17}ClN_2$)Cl$_3$]–1445.2, [Cu($C_{22}H_{17}ClN_2$)$_2$Cl$_2$]·5H$_2$O–1447.2, [Cu($C_{22}H_{17}ClN_2$)$_4$] (C$_6$H$_4$NO$_2$)$_2$–1449.4 cm$^-$. Another stretching band of 1489.9 cm$^-$ of the clotrimazole C=N bond is less intensive and its position is practically unchanged in the salts. In the mono- and biligand salts of the metal ions, there are no absorption bands of undissociated carboxyl groups of aromatic acids: HBenz is 1688, HNic is 1701, and H$_2$Sal is 1655 cm$^-$. However, in the range of 1540–1360 cm$^{-1}$, there are bands that are responsible for asymmetric and symmetric valence vibrations of COO$^-$ groups.

In our work, the conditions for synthesising the mixed-ligand salt of the composition [Cu($C_{22}H_{17}ClN_2$)$_2$Cl$_2$]·5H$_2$O (CuCl$_2$:Clm = 1:2, ethanol, aqueous ethanol solutions, room temperature) differ from the conditions of the synthesis described by the authors in [8]. When boiling the mixture of the solid copper salt and the ethanol solution of clotrimazole (M:L = 1:3) for 4 h involving a reflux condenser, the authors sequentially isolated two salts of the composition [Cu(Clm)$_2$Cl$_2$·5H$_2$O and Cu(Clm)$_2$EtOHCl$_2$. Our synthesis method and the salt of the composition [Ag($C_{22}H_{17}ClN_2$)$_2$]NO$_3$·2H$_2$O differ from those described in [4]; the [Ag($C_{22}H_{17}ClN_2$)$_2$]NO$_3$ salt was isolated from ethanol solutions by the authors.

Clotrimazole acts as a monodentate ligand in the formation of mixed-ligand copper(II) salts [Cu($C_{22}H_{17}ClN_2$)$_4$](C$_6$H$_4$NO$_2$)$_2$, [Cu($C_{22}H_{17}ClN_2$)$_4$](C$_7$H$_5$O$_2$)$_2$, Cu($C_{22}H_{17}ClN_2$)$_3$ (C$_7$H$_4$O$_3$)·2H$_2$O. The acid anion can also enter the inner sphere of the complexing agent. This fact can be attributed to the affinity of the Cu$^{2+}$ ($d^9$) ion to the donor nitrogen atoms of clotrimazole and to the donor oxygen atoms of aromatic acids. It is associated with the presence of the mutual influence of ligands in the inner sphere due to the $\pi$-$\pi$-interaction of rings of aromatic acids and azole and the formation of hydrogen bonds, including different acid–base natures of ligands.

Clotrimazole is an antimycotic agent, used to treat mycosis. Figure 1 allows for the conclusion that the difference in the diameters of halos that inhibit the growth and development of *Saccharomyces cerevisiae* is conditioned by the greater antimycotic activity of the $[Ag(C_{22}H_{17}ClN_2)_2]NO_3 \cdot 2H_2O$ complex as compared to that of clotrimazole and silver nitrate (statistically confirmed, $\rho < 0.001$). This means that there is synergism in the action of silver ions and clotrimazole.

Cocrystals/salts represent crystal phases consisting of two or more different molecular and/or ionic, usually stoichiometric, compounds. Intermolecular hydrogen bonds, van der Waals and electrostatic interactions, and $\pi$-interactions play an essential role in the formation of such compounds. Cocrystals/salts of clotrimazole containing nicotinic, benzoic, and salicylic acids were synthesised when the pH was ~4.5 and the folic acid was 6. Table 4 shows that the mole ratio of $Clm:H_{(1-2)}L$ in the synthesised compounds is compliant with the monobasicity of the clotrimazole weak base and the basicity of HNic, HBenz, and $H_2Sal$ acids. An $H_3Fol$ folic acid molecule represents a tribasic acid yielding sequentially two separate protons of carboxylic groups of the glutamic residue. Additionally, the proton detaches from the $NH_3^+$ group of the zwitter-ion only when the pH > 8. The diagram of the folic acid particle yield (Figure 4) shows that when the pH is ~6, the acid must act like dibasic acid. In fact, we have demonstrated that in the case of a stronger base of imidazole ($lgB_1 = 7.69$) than that of clotrimazole (($lgB_1 = 5.99$), the mole ratio of $Im:H_3Fol$ in the product $3Im \cdot H_2Fol \cdot 2H_2O$ of their interaction (synthesis pH is ~7) is 3:1 [28]. Additionally, the mole ratio of $Clm:H_2Fol$ in the product of the $2Clm \cdot H_2Fol \cdot H_2O$ interaction (synthesis pH is ~6) is 2:1 (Table 4).

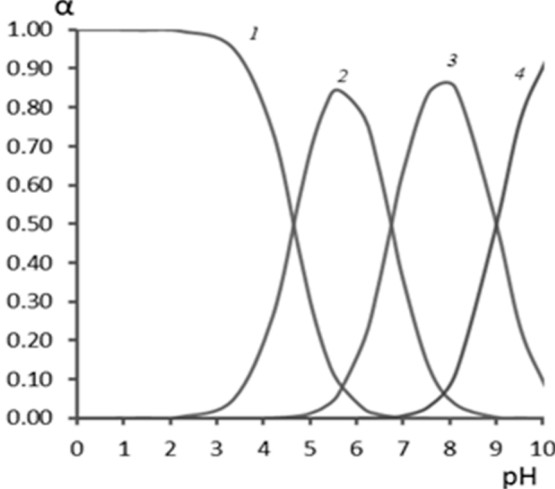

**Figure 4.** Diagram of the folic acid particle yield: $1-H_3Fol$, $2-H_2Fol^-$, $3-HFol^{2-}$, $4-Fol^{3-}$ ($C$ ($H_3Fol$) = 0.01 mol/L; $lgB_1$ = 9.00, $lgB_2$ = 15.75, $lgB_3$ = 20.40; $pK_1$ = 4.65, $pK_2$ = 6.75, $pK_3$ = 9.00).

In the case of a cocrystal/salt of $2C_{22}H_{17}ClN_2 \cdot C_7H_6O_3$ (Table 4), the salicylic acid when reacting with clotrimazole acts as bicarboxylic acid. In contrast to derivatives of aliphatic dicarboxylic acids, all ionic salts of aromatic dicarboxylic acids were obtained only in the form of medium salts, despite the discovered empirical dependence ($\Delta pK_{a(2-1)} > 2$), allowing the existence of both acidic and medium salts of aromatic dicarboxylic acids [29].

The thermal behaviour of the synthesised cocrystals/salts was analysed by means of thermogravimetric analysis. The presence of endoeffects when the sample mass does not change in the thermograms of $Clm \cdot HNic$ (Figure 5) and $2Clm \cdot H_2Sal$ allows for the determination of their melting temperatures of 119.3 and 114.4 °C, respectively. These temperatures are lower than the melting temperatures of the initial components (148, 237, and 159 °C for Clm, HNic, and $H_2Sal$). This fact can be an indirect confirmation of the formation of cocrystals/salts since the mixtures of two substances (especially of the eutectic type) are known to frequently crystallise at a lower temperature than that of the individual components. The confirmation of the formation of cocrystals/salts

consists of the fact that their thermal decomposition proceeds along with the elimination of the initial components: clotrimazole gaining the endoeffect in the temperature range of ~(200–400) °C; acids gaining the exoeffect at ~(400–800) °C (a thermal decomposition scheme of the cocrystal/salt Clm·HNic is provided):

$$\text{Clm·HNic} \xrightarrow[\text{Endoef., 199–400 °C}]{-\text{Clm(f.72.2\%; c.73.69\%)}} > \text{HNic} \xrightarrow[\text{Exoef., 400–800 °C}]{-\text{HNic(f.27.8\%; c.26.29\%)}}.$$

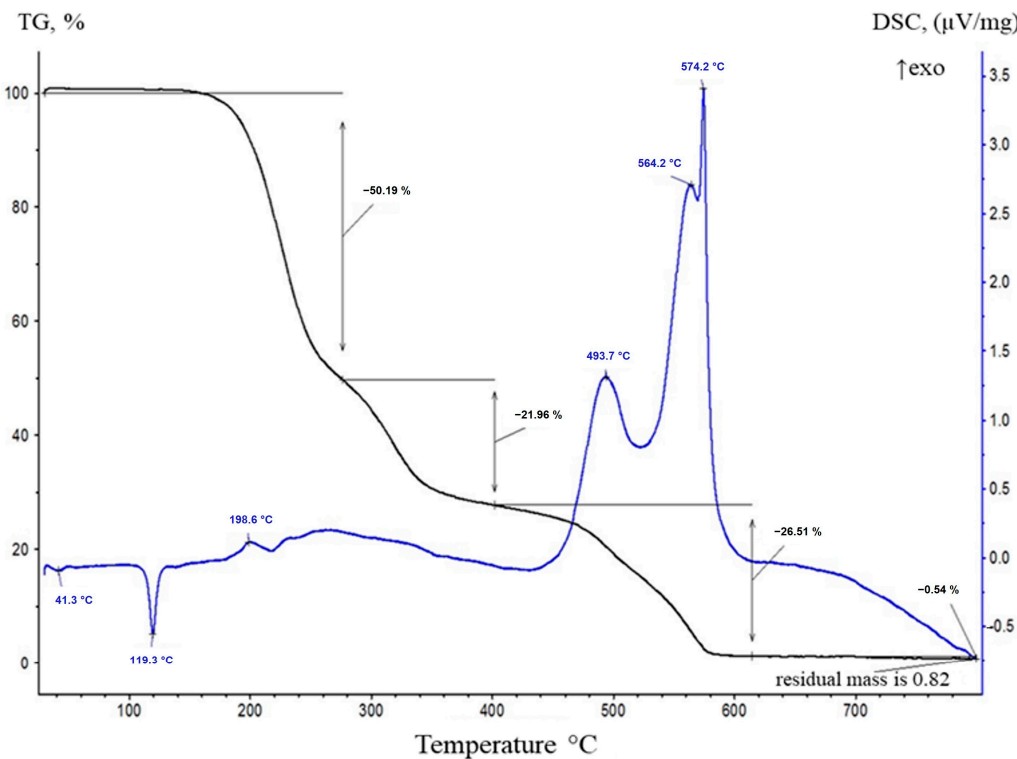

**Figure 5.** TG and DSC curves of the $C_{22}H_{17}ClN_2 \cdot C_6H_4NO_2$ cocrystal/salt (in the air).

Analysis of the IR spectra of the synthesised cocrystals/salts has shown that the absorption band of the nondissociated carboxylic group of the salicylic acid (1655 cm$^-$) is absent from the IR spectrum of the 2Clm·H$_2$Sal compound, but there are bands belonging to asymmetrical and symmetrical valence vibrations of COO$^-$ groups (in the range of 1600–1400 cm$^-$). The stretching band of the clotrimazole heterocycle bond C=N (1436.9 cm$^-$) in this compound is shifted to the high-frequency region (1464.8 cm$^-$). The absorption band of the nondissociated carboxylic group (1688 cm$^-$) is also absent from the IR spectrum of the Clm·HBenz compound. The absorption bands (1694.4 cm$^-$, 1701 cm$^-$) of the nondissociated carboxylic group and the bands of asymmetrical and symmetrical valence vibrations of COO$^-$ groups are present in the 2Clm·H$_2$Fol and Clm·HNic compounds in IR spectra. The position of the Clm band of 1489.9 cm$^-$ is practically unchanged in the mentioned compounds (1484.9, 1489.8, 1482.4, and 1490.7 cm$^-$, respectively). The band (1080.6 cm$^-$) of deformation vibrations of cyclic C–H bonds remains unchanged in cocrystals/salts.

The IR spectroscopy data on the cocrystals/salts (the presence of the absorption bands of nondissociated and dissociated carboxylic groups of HNic and H$_2$Fol acids and the absence of absorption bands of nondissociated carboxylic groups of HBenz and H$_2$Sal) are in good agreement with the change (increase) in the value $\Delta pK_a = pK_a$ (the base is Clm, 5.99),

| | Acid | HNic | H$_2$Fol | HBenz | H$_2$Sal |
|---|---|---|---|---|---|
| $pK_a$ (acid): | $pK_{a1}$ | 4.84 | 4.65 | 4.01 | 2.83 |
| | $\Delta pK_{a1}$ | 1.15 | 1.34 | 1.98 | 3.16 |

An acid strength increase by the first stage of dissociation is observed among the HNic, $H_2Fol$, HBenz, and $H_2Sal$ acids. The difference between the clotrimazole $pK_a$ value and the first constant of dissociation of nicotinic, folic acids implies that the Clm–HNic, Clm–$H_2Fol$ combinations will result in the formation of the cocrystals or a cocrystallised mixture of substances. According to Table 2, this fact is confirmed by the data of their thermal decomposition and by the data of the diffraction pattern of the Clm–HNic compound, where new reflexes ($2\theta° = 12.5; 18.42; 31.84; 34.6$) appear, differing from the reflexes of the initial components. The binary 2Clm–$H_2Fol$ system can be considered a cocrystallised mixture of substances since there are no new reflexes in the diffraction pattern of this system that differ from the initial components; i.e., the crystallographic identity of the initial components is preserved in the synthesis product (Figure 6).

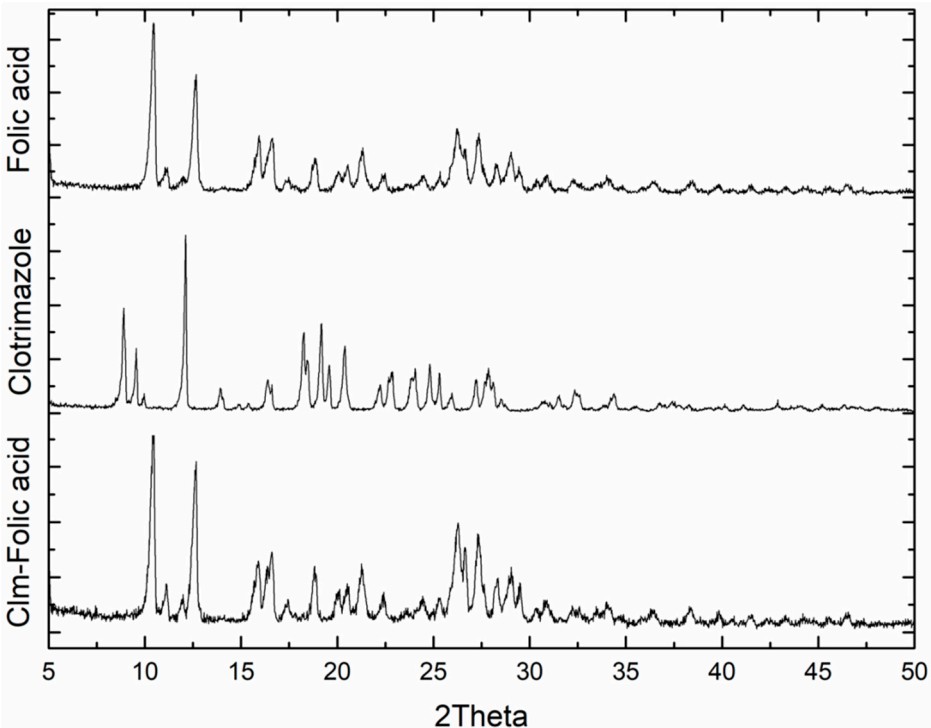

**Figure 6.** Diffraction patterns of powders of folic acid, clotrimazole, and the binary 2Clm–$H_2Fol$ system; (new reflexes are absent from the binary system).

Combinations such as Clm–HBenz (new reflexes are $2\theta° = 12.48; 19.44; 25.15; 34.83$) and 2Clm–$H_2Sal$ (new reflexes are $2\theta° = 12.48; 19.52; 20.93$, Figure 7) must be in the range of a cocrystal–salt continuum by the value of $\Delta pK_{a1}$. This means that in these systems, a partial or complete transfer of the proton is possible from the acid to clotrimazole, accompanied by the formation of clotrimazolium salts (although the molecule ionization condition in the crystal is often unpredictable). The difference between a crystal salt and a cocrystal consists of the degree of proton transfer. Depending on the organic base strength, the compounds represent either ionic salts, where the nitrogen atoms of the aromatic ring have been protonated, or supermolecular complexes, where the organic acid protons participate in the formation of hydrogen bonds.

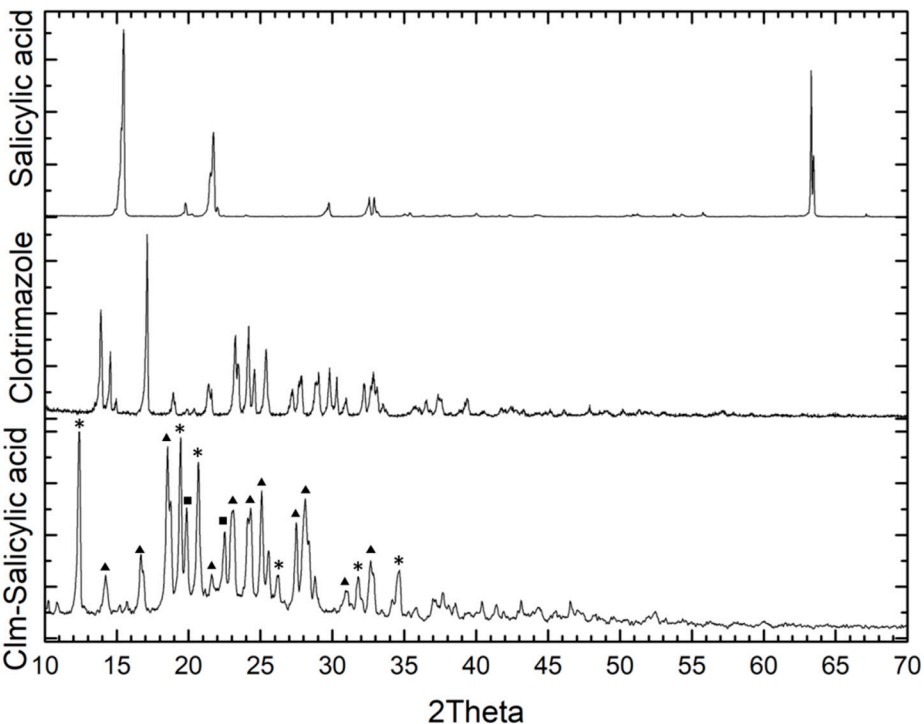

**Figure 7.** Diffraction patterns of the powders of the binary 2Clm–$H_2$Sal system and the initial components. Symbols: * represents peaks that are absent from the diffraction patterns of the initial substances; ■ and ▲ represent peaks that correspond to salicylic acid and clotrimazole.

## 5. Conclusions

1.  The conditions (molar ratio of the components, solvent type, pH of ~(5.0–5.5)) for synthesising mixed-ligand salts from the $AgNO_3$, $H[AuCl_4]$, and $CuCl_2$ solutions, as well as from aqueous suspensions of slightly soluble copper(II) salts containing nicotinate, benzoate, and salicylate anions, including the ethanol clotrimazole solution, were found and substantiated. The isolated compounds were characterised by elemental, thermal, thermogravimetric, and IR spectroscopic analyses.

2.  The novelty of this work consists of the fact that MLS of $AgNO_3$, $H[AuCl_4]$, and $CuCl_2$ containing clotrimazole were obtained by methods that are different from those described in the literature. New MLS of copper(II), having clotrimazole and anions of nicotinic, benzoic, and salicylic acids, as well as clotrimazole cocrystals/salts containing the mentioned aromatic acids, were produced.

3.  The $[Ag(Clm)_2]NO_3 \cdot 2H_2O$ compound was shown to have a higher antifungal activity than that of the initial $AgNO_3$ and $C_{22}H_{17}ClN_2$.

4.  The electronic spectra of absorbing alcoholic solutions, containing copper(II) chloride and Clm, have a higher optical density compared to that of the solutions of the individual components of the same concentration, which indicates the formation of a new compound in the $CuCl_2$–Clm system: a mixed-ligand complex.

5.  Cocrystals/salts having a molar ratio of the components equal to 1:1, 1:1, 2:1, and 2:1 were obtained from the aqueous or aqueous ethanol suspensions containing clotrimazole and, respectively, nicotinic, benzoic, salicylic, or folic acids in the pH range of ~(4.5–6.0). These cocrystals/salts were subject to thermogravimetric and IR spectroscopic studies. Diffraction patterns of the powders were obtained. The influence of the difference in the $pK_a$ components on the ability to form cocrystals/salts was estimated. In the above-mentioned compounds, organic acids and clotrimazole exhibit their characteristic acid–base properties.

6.  The formation of clotrimazole crystals/salts involving nicotinic, benzoic, and salicylic acids can serve as indirect evidence of the compatibility of these substances in the

inner sphere of the metal complex in the solution and in individual compounds. Such studies can be used to plan the synthesis of mixed-ligand compounds with *d*-metal ions.

**Author Contributions:** N.S.: conceptualization, chemical synthesis, methodology, research, formal analysis, data curation, validation. I.K.: methodology, formal analysis, validation, data curation, writing the manuscript, review and editing. V.K. (Vladislav Korostelev): research, formal analysis. D.F.: research, formal analysis. V.K. (Vladimir Kozik): data curation, review and editing. All authors have read and agreed to the published version of the manuscript.

**Funding:** The reported study was funded by Tomsk State University Development Program (Priority-2030).

**Data Availability Statement:** Not applicable.

**Conflicts of Interest:** The authors declare no conflict of interest. The funding sponsors had no role in the design of the study; in the collection, analysis, or interpretation of data; in the writing of the manuscript; and in the decision to publish the results.

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
