# Peer review of "Interactions of Clotrimazole with Certain d-Metal Compounds and with Organic Acids"

_inorganics, doi:10.3390/inorganics11100393_

Round 1

Reviewer 1 Report (Previous Reviewer 3)

General comment

The author presents a manuscript describing the synthesis and characterization of clotrimazole complexes and biological evaluations. I believe that the paper lacks depth in describing the motivation for these particular compounds and the bioactivity. I recommend publication after the authors address the following issues:

Specifical comments:

1) The results displayed in this paper are nevertheless interesting but need to be better organized to facilitate understanding. Also, grammatical errors of the English need to be corrected here and there.

2) The abstract should be reformulated in order to make it clearer what was done and the results obtained in work.

3) The author should better organize the synthetic routes used to facilitate visualization;

4) Two representations are used for the complexes: [Ag(C22H17ClN2)2]NO3 and [AgClm2]NO3. The same occurs to copper complexes. This should be standardized.

5) The experimental section should be reorganized. Some results should be discussed in the results section and not in the experimental section, such as antimycotic activity.

6) Mass spectrometry should be performed for the studied complexes.

7) A scheme with the structures of the complexes obtained would facilitate visualization. This could be provided in a more elaborate synthetic route.  

8) Were coordination complexes or co-crystals obtained? This got confused in the text.

Author Response

Reviewer 2 Report (New Reviewer)

This article is described about the clotrimazole complexes with d-metals or organic acids. These experiments are appropriate enough to explain the author's suggestions, I think. Therefore, I agree the manuscript is published, after checking as shown below.

The unit of mole concentration should be described "mol/L" not "mole/l".

The products should be described ”x" not "·"  (e.g. 5x10-6) 

In page 11, I can hardly understand the the expression of inequality(?) on the bottom line. More clear expression wants to be carried out. 

Diffraction pattern of Clm-HBenz was not shown, only the angles are described in the manuscript. There had also better be the figure of diffraction pattern of Clm-HBenz, I think.

Author Response

Reviewer 3 Report (New Reviewer)

The scientific content of the ms. by Karzina and co-workers, is interesting and I favor its acceptance, albeit with revisions. The paper may attract the interest of scientists working in the areas of the coordination chemistry of azoles, the use of silver(I) compounds as antimicrobial agents and the structural chemistry of cocrystals, although the latter subject does not strictly belong to inorganic chemistry. Also, I hope that the article will receive a number of citations in the future. Features of this work – which justify my proposal for acceptance – are: (a) The azole ligand is interesting and promising for further coordination chemistry, but mainly metallosupramolecular chemistry. (b) The silver(I) complex shows very good antimycotic activity towards a fungi strain; and (c) The cocrystals formed are interesting. Disadvantages of this work are: (i) The lack of single-crystal X-ray structures for representative compounds, and  (ii) the non-employment of NMR techniques for better characterization of the present compounds. The quality of schemes/figures is good and the references list covers the topic under study satisfactorily.

Major revision points/comments/suggestions to be taken into account by the author:

(1) The English of the ms should be improved.

(2) The ms is long and should be drastically condensed.

(3) Section 5, “Conclusion”: The perspectives of the present work should be outlined.

(4) Some thermal decomposition data should be refined. For example in Table 2: The benzoate anion can not be removed alone! Probably neutral benzoic acid is lost!

(5) The coordination entities of the complexes should be written within parentheses, i.e. [Au(Clm)Cl3], [Ag(Clm)2](NO3),…

(6) I prefer the term “silver” and “gold” instead of “argentum” and “aurum”, respectively.

(7) The IR discussion is incomplete. For example, did the authors locate the nitrate bands in the spectrum of the Ag(I) complex? From the frequencies and numbers of these bands, they can decide if the nitrate is ionic (as they propose) or coordinated resulting in a 3-coordinate complex.

(8) I advise the authors to prepare a new chart/scheme illustrating (in a schematic way, e.g. using ChemDraw) the proposed molecular structures of the binary and ternary (mixed-ligand) copper(II) complexes.

(9) All the formulae of the complexes (binary and ternary) and cocrystals should be written using the abbreviation of the ligands, e.g. [Cu(Clm)2Cl2]∙5H2O, [Cu(Clm)4(nic)2] (nic- is the nicotinate ligand), [Cu(Clm)3(sal)] (sal2- is the dianion of salicylic acid, H2sal), Clam∙Hnic, etc. … In the ms the authors use two ways of writing the formulae, the other is with the numbers of C, H, N and O atoms, e.g. Cu(C12H17ClN2)3(C7H4O3) etc. … This dual writing is confusing.

(10) Table 1: The final residue of the Ag(I) and Au(III) complexes cannot be CuO!!!

The English of the ms need corrections.

Author Response

Reviewer 4 Report (New Reviewer)

This paper dealt with preparation and pH-dependent properties of Ag and Cu complexes with imidazole-related ligand toward biological activity.

Functional studies and their results may be significant (if reliable), characterization of complexes was weak and rough (mainly elemental analysis ~some compounds were not in agreement with capable deviation~ and IR spectroscopy). May impurity exist in the systems investigated? The premise of the compound lacks credibility, so it should be reinforced.

That's all.

Round 2

Reviewer 1 Report (Previous Reviewer 3)

The authors did not take into consideration my suggestions regarding the placements. Nonetheless, I believe the manuscript has shown improvement and is suitable for acceptance in its current form for publication.

Author Response

Thank you very much. 

Reviewer 3 Report (New Reviewer)

I am not satisfied with the work that the authors have done during the revision of the ms. Many of my revision points/comments/suggestions have not been taken into account. Also, I see that the authors have not addressed satisfactorily many points raised by reviewers 1 and 4. For this reason I suggest another cycle of revision or rejection. 

The English of the ms are good in general terms.

Author Response

The authors do not agree with the respected reviewer that many of his improvements/comments/suggestions were not taken into account. The comments (1, 3, 4, 5, 6, 10) were taken into account, the corresponding corrections and additions were made in the text. Controversial remarks (2, 9), with the permission of the reviewer, are resolved taking into account the opinion of the authors. The reviewer's suggestions (7, 8) cannot be implemented by the authors, due to the complexity of additional research and refinement. We believe that in the presented form, the article answers the main objectives of the study. However, if the reviewer insists on the correction and revision of comments 7 and 8, we ask for additional time for correction.

Round 3

Reviewer 3 Report (New Reviewer)

I see that the other three(3) colleagues have commented warmly on the original and revised versions of the manuscript. Since I am a minority member of the reviewers panel, I have to agree with the opinions of the other reviewers. Generally, I believe that the papers are authors’ papers (and not reviewers’ papers)! I see that the authors insist on presenting the complexes with both their gross and analytical formulae, please see my comment no. 9 in my first review. According to my opinion, this dual writing is confusing for the readers and not a common practice in Coordination Chemistry papers. However, if this is ok for the authors, I do not have any problem! Thus, I propose acceptance either the authors agree with my opinion or not.

I see that the other three(3) colleagues have commented warmly on the original and revised versions of the manuscript. Since I am a minority member of the reviewers panel, I have to agree with the opinions of the other reviewers. Generally, I believe that the papers are authors’ papers (and not reviewers’ papers)! I see that the authors insist on presenting the complexes with both their gross and analytical formulae, please see my comment no. 9 in my first review. According to my opinion, this dual writing is confusing for the readers and not a common practice in Coordination Chemistry papers. However, if this is ok for the authors, I do not have any problem! Thus, I propose acceptance either the authors agree with my opinion or not.

Author Response

The authors find that representing complexes both as gross formulas and as analytic formulas is acceptable.

This manuscript is a resubmission of an earlier submission. The following is a list of the peer review reports and author responses from that submission.

Round 1

Reviewer 1 Report

This manuscript can be accepted for publication in INORGANICS with some minor English language corrections and using the most familiar words, for example in chemistry literature we use deprotonation instead of decolonization!

Reviewer 2 Report

  In the manuscript “Interaction between clotrimazole and compounds of some d metals and organic acids’’, the author presented synthesis of the complexes between different Ag(I), Au(III), Cu(II) salts and clotrimazole. The obtained complexes are characterized by IR spectroscopy, elemental, thermal and thermogravimetric analysis. Also, the antimicrobial activity of the [Ag(C22H17ClN2)2]NO3 ∙ 2H2O complex is analyzed against the Saccharomyces cerevisiae.

The synthesis of cocrystal salts are preformed and analyzed as well .

The manuscript is not well organized and connections between the sections are missing. Conclusions are very general, and discussion is unclear with too many assumptions. Additionally, manuscript lacks a lot of novelty where the similar complexes have been reported before. Some other issues need to be addressed as well.

1.      More spectroscopic techniques are needed for the characterization of the obtained complexes.

2.      In the introduction part of the manuscript the authors described the antimicrobial activity of the similar complexes with the same ligand, but however in their work the antimicrobial activity is analyzed just for one complex. Antimicrobial activity of the other analyzed complexes is required and comparison with literature results.

3.      The author used well diffusion method for determination of antifungal activity. Dilution method is more recommended, since its more accurate.  

    Overall I do not recommend this manuscript for the publication in Inorganics.

Reviewer 3 Report

General comment

The author presents a manuscript describing the synthesis and characterization of clotrimazole complexes and biological evaluations. I believe that the paper lacks depth in describing the motivation for these particular compounds and the bioactivity. I recommend publication after the authors address the following issues:

Specifical comments:

1) The results displayed in this paper are nevertheless interesting but need to be better organized to facilitate understanding. Also, grammatical errors of the English need to be corrected here and there.

2) The abstract should be reformulated in order to make it clearer what was done and the results obtained in work.

3) The author should better organize the synthetic routes used to facilitate visualization;

4) Two representations are used for the complexes: [Ag(C22H17ClN2)2]NO3 and [AgClm2]NO3. The same occurs to copper complexes. This should be standardized.

5) The experimental section should be reorganized. Some results should be discussed in the results section and not in the experimental section, such as antimycotic activity.

6) Mass spectrometry should be performed for the studied complexes.

7) A scheme with the structures of the complexes obtained would facilitate visualization. This could be provided in a more elaborate synthetic route.  

8) Were coordination complexes or co-crystals obtained? This got confused in the text.